# Nanowire Neural Networks for time-series processing

**Veronica Pistolesi**
Dept. of Computer Science
University of Pisa, Italy
v.pistolesi6@studenti.unipi.it

**Andrea Ceni**
Dept. of Computer Science
University of Pisa, Italy
andrea.ceni@di.unipi.it

**Gianluca Milano**
AM&D-Group
INRiM, Torino, Italy
g.milano@inrim.it

**Carlo Ricciardi**
Dept. of Applied Science and Technology
Politecnico di Torino, Italy
carlo.ricciardi@polito.it

**Claudio Gallicchio**
Dept. of Computer Science
University of Pisa, Italy
claudio.gallicchio@unipi.it

## Abstract

We introduce a novel computational framework inspired by the physics of nanowire memristive networks, which we embed into the context of Recurrent Neural Networks (RNNs) for time-series processing. Our proposed Nanowire Neural Network architecture leverages both the principles of Reservoir Computing (RC) and fully trainable RNNs, providing a versatile platform for sequence learning.

We demonstrate the effectiveness of the proposed approach across diverse regression and classification tasks, showcasing performance that is competitive with traditional RC and fully trainable RNNs. Our results highlight the scalability and adaptability of nanowire-based architectures, offering a promising path toward efficient neuromorphic computing for complex sequence-based applications.

## 1 Introduction

Neuromorphic computing, inspired by the structure and function of the human brain, aims to replicate biological information processing systems for the development of efficient and brain-like computational frameworks [Marković et al., 2020]. A key approach within this field is leveraging physical substrates, which bypass traditional digital simulations, to perform computation based on their inherent physical properties [Nakajima, 2020]. Such systems offer an opportunity to implement more scalable and efficient computational models compared to conventional methods.

Reservoir Computing (RC), and particularly the Echo State Network (ESN) approach, provides a theoretical framework that aligns well with these principles, particularly for recurrent neural networks (RNNs) implemented in physical systems [Nakajima and Fischer, 2021, Yan et al., 2024]. In RC, the recurrent layer, known as the reservoir, remains fixed during training, while only the readout layer is trained. This reduces training complexity and allows efficient mapping of input signals into high-dimensional spaces for tasks such as time-series processing. The fixed nature of the reservoir offers a natural avenue for integrating physical systems, enabling their use as computational reservoirs [Tanaka et al., 2019].

Second Workshop on Machine Learning with New Compute Paradigms at NeurIPS 2024 (MLNCP 2024).

Recent work by Milano et al. [Milano et al., 2022] takes this concept further by demonstrating how self-organizing memristive nanowire networks can serve as physical reservoirs for computation. These nanowire networks map spatio-temporal inputs into a feature space through their inherent nonlinear dynamics, offering fault tolerance and adaptability, akin to biological neuronal circuits. This approach, termed in-materia computing, underscores the potential for neuromorphic architectures to implement brain-inspired paradigms with reduced training costs while maintaining robust performance in tasks like time-series prediction.

Building on this foundation, we propose to explore the scalability and potential of nanowire networks by introducing physics-inspired computational models, named Nanowire Echo State Network (NW-ESN), and Nanowire Recurrent Neural Network (NW-RNN). We analyze the architectural parameters of these models and assess their performance across several time-series tasks. To validate our approach, we perform experiments on both classification and regression tasks. Through these experiments, we demonstrate that nanowire-based models can achieve competitive results compared to traditional methods, providing insights into their scalability and applicability in neuromorphic computing. Our findings highlight the potential of these models in advancing next-generation intelligent systems and robust, adaptive computational frameworks.

## 2 Nanowire networks

Nanowire networks consist of tiny, highly conductive wires, made from metallic materials, arranged in a mesh-like pattern, see Fig.1 for a realistic picture. These networks mimic the physical structure of the human brain, with the nanowires acting like neurons and their intersections resembling synapses. This unique structure allows nanowire networks to exhibit properties similar to neural networks, including the ability to learn and remember. Interestingly, the single memristive dynamics

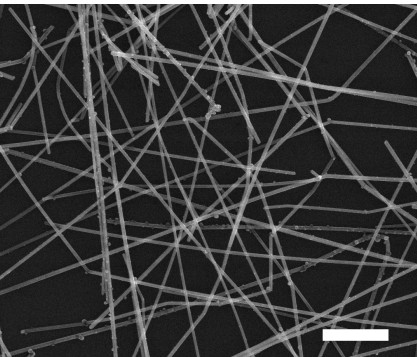

Figure 1: Image of a nanoscale self-assembled Ag nanowire network obtained through scanning electron microscopy (scale bar, 2 $\mu m$). This represents a memristive network, since each junction between nanowires is a memristive element and the current flow relies on the history of applied voltage and current.

at the intersection of two intersecting nanowires can be modeled as the transfer of ions through the membrane of a neuronal cell, as is the case for the three ionic channels of the Hodgkin–Huxley model [Nelson and Rinzel, 1995]. Specifically, the state equation for the synaptic properties of the NW under electrical stimulation is expressed as the following potentiation–depression rate balance equation Miranda et al. [2020]:

$$\frac{dg_{ij}}{dt} = \kappa_{P,ij}(V_{ij}) \cdot (1 - g_{ij}) - \kappa_{D,ij}(V_{ij}) \cdot g_{ij} \tag{1}$$

In Eq. 1, $g_{ij}$ is the normalized conductance (memory state) that assume values in between 0 and 1 while $\kappa_{P,ij}(V_{ij})$ and $\kappa_{D,ij}(V_{ij})$ are the potentiation and depression rate coefficients that are assumed to be function of the applied voltage through exponential relations, as expected for diffusion of ions:

$$\kappa_{P,ij}(V_{ij}) = \kappa_{P0} exp(+\eta_P V_{ij}) \tag{2}$$

$$\kappa_{D,ij}(V_{ij}) = \kappa_{D0} exp(-\eta_D V_{ij}), \tag{3}$$

where $\kappa_{P0}$, $\kappa_{D0} > 0$ are constants while $\eta_P$, $\eta_D > 0$ are transition rates.

# 3 Computational neural model

In this work, we develop Nanowire Neural Networks for time-series processing where the physical memory state $g$ becomes the nanowire network state. To achieve this, we rewrite Eq. 1, 2 and 3 as:

$$\frac{dh}{dt} = K_p(x) \cdot (1 - h) - K_d(x) \cdot h, \tag{4}$$

$$K_p = K_{p_0} \cdot e^{\eta_p x}, \qquad K_d = K_{d_0} \cdot e^{-\eta_d x}, \tag{5}$$

where $0 \leq h \leq 1$ is the reservoir state and $K_p$ and $K_d$ are the two non-linearities which process the input $x$, keeping the other physical variables ($K_{p0}$, $K_{d0}$, $\eta_p$, $\eta_d > 0$) as network hyperparameters. The state transition function (Eq.6) of a single Nanowire Neuron is obtained by discretizing Eq. 4 using the forward Euler's method, leading to the following formulation:

$$h(t + 1) = h(t) + \Delta_t[K_p(x(t + 1)) - (K_p(x(t + 1)) + K_d(x(t + 1)))h(t)]. \tag{6}$$

Equation 6 is then used to construct the fully connected Nanowire Neural Network (Fig.2).

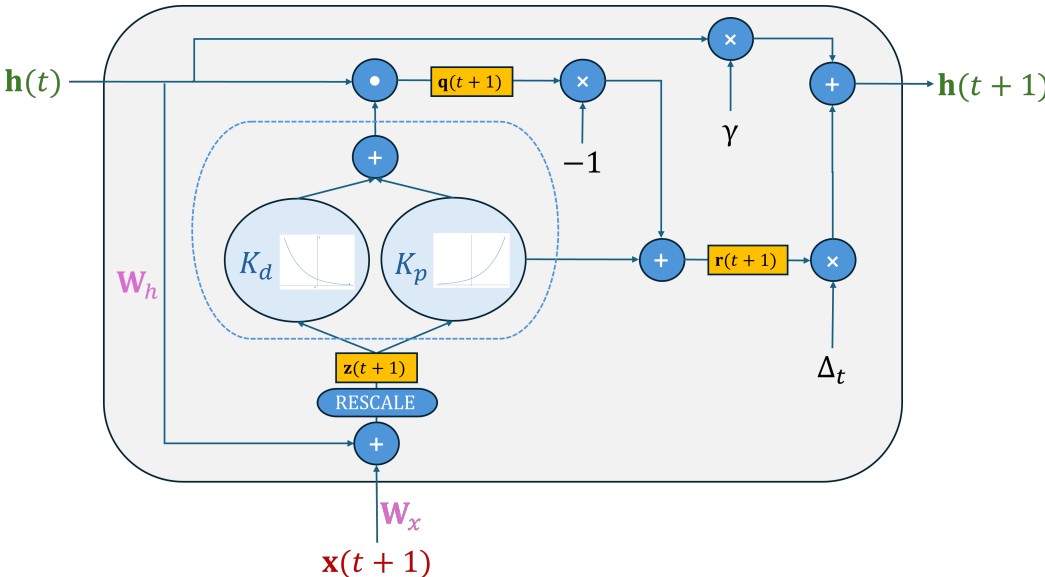

Figure 2: Nanowire Network computational graph where: $\mathbf{h}(t)$ and $\mathbf{h}(t+1)$, in green, are the current and the next hidden state vectors, respectively, $\mathbf{x}(t+1)$, in red, is the external input vector, $\mathbf{W}_h$ is the recurrent weight matrix (or recurrent kernel), $\mathbf{W}_x$, in magenta, is the input weight matrix (or kernel) and $\Delta_t$ is the discretization time (non-linearity operations and their trajectories are marked by a dashed line).

The dynamics of the Nanowire Neural Network are described by the following set of equations:

$$\mathbf{z}(t + 1) = \text{RESCALE}\big(\mathbf{W}_h\mathbf{h}(t) + \mathbf{W}_x\mathbf{x}(t + 1) + \mathbf{b}\big), \tag{7}$$
$$\mathbf{q}(t + 1) = diag(K_p(\mathbf{z}(t + 1) + K_d(\mathbf{z}(t + 1))\mathbf{h}(t), \tag{8}$$
$$\mathbf{r}(t + 1) = K_p(\mathbf{z}(t + 1)) - \mathbf{q}(t + 1), \tag{9}$$
$$\mathbf{h}(t + 1) = \mathbf{r}(t + 1)\Delta_t + \gamma\mathbf{h}(t), \tag{10}$$

where $\mathbf{h}(t)$ indicates the state of the recurrent layer, $\mathbf{x}(t)$ is the external input signal, $\mathbf{W}_h$ is the recurrent kernel, $\mathbf{W}_x$ is the (input-to-recurrent) kernel, and $\mathbf{b}$ is a bias vector.

Our proposed model admits the following fundamental set of (nanowire-related) hyperparameters: $K_{p0}$, $K_{d0}$, $\eta_p$, $\eta_d$, $\Delta_t$, and $\gamma$. Notice that we introduce the $\gamma$ factor to allow the model to show asymptotically stable dynamics (the Echo State Property [Yildiz et al., 2012]) when $\gamma < 1$. Moreover, for a certain configuration of hyperparameters, we observe that manipulating the input $\mathbf{x}$ impacts the reservoir state. Notably, a specific input range allows system's fixed point of Eq. 4 to move in a suitable region for information representation and learning. For this reason, the result of the affine

transformation in Eq. 7 is subject to a proper rescaling operation before entering in the non-linearities. This heuristic enhances the stability and performance of the computational model Eq.s 7-10, and is computed component-wise as follows:

$$\text{RESCALE}(y) = \frac{(b-a)}{1 + \exp\left(-y \cdot s\right)} + a, \qquad (11)$$

where $a = 0.35$, $b = 1.15$, and $s$ is a slope parameter.

The computational graph of the proposed Nanowire Neural Network is illustrated in Figure 2.

In the following, we give two specific instances of the approach introduced so far, by framing it in the context of RC and fully trainable RNNs.

### 3.1 NW-ESN: Reservoir Model

NW-ESN is the reservoir-based Nanowire Neural Network Model in which both the kernel ($\mathbf{W}_x$) and the recurrent kernel ($\mathbf{W}_h$) matrices are initialized under stability constraints and left untrained. As usual in RC applications, we rescale the kernel $\mathbf{W}_x$, the bias $\mathbf{b}$, and recurrent kernel $\mathbf{W}_h$, with the input scaling ($\omega$), the bias scaling ($\beta$), and the spectral radius ($\rho$), respectively, and we use them as hyperparameters of the model.

### 3.2 NW-RNN: Fully trainable model

NW-RNN is the fully trainable version of the Nanowire Neural Network Model where both the kernel ($\mathbf{W}_x$) and the recurrent kernel ($\mathbf{W}_h$) matrices are trained end-to-end with backpropagation through time (BPTT).

## 4 Experiments

We have validated our models for both time series classification and regression problems. To this end, we have considered the following datasets: `FordA`, `ECG5000`, `SyntheticControl` and `EarthQuakes` from Dau et al. [2019] for the classification task; `FloodModeling1` and `FloodModeling2` from Tan et al. [2021] for the regression task. A synthetic description of the datasets properties can be found in Table 1.

| Dataset | Train Size | Test Size | TS Length |
|---|---|---|---|
| FordA Dau et al. [2019] | 3601 | 1320 | 500 |
| ECG5000 Dau et al. [2019] | 500 | 4500 | 140 |
| SyntheticControl Dau et al. [2019] | 300 | 300 | 60 |
| Earthquakes Dau et al. [2019] | 322 | 139 | 512 |
| FloodModeling1 Tan et al. [2021] | 471 | 202 | 266 |
| FloodModeling2 Tan et al. [2021] | 389 | 167 | 266 |

Table 1: Datasets overview.

All datasets were provided with a predefined split into training and test sets. We further divided the training set to obtain a development set, using $80\%$ of data for training and $20\%$ for validation. This setup was used consistently across all experiments for an initial model selection phase through grid search, followed by a model assessment phase.

In all experiments, the performance achieved by our models NW-RNN and NW-ESN was compared with that of conventional RNNs and ESNs. Specifically, for the ESN model, we have considered the leaky-integrator neuron formulation [Jaeger et al., 2007], which includes a leaking-rate hyperparameter $\alpha$. For all experiments, we used networks with a number of $N = 100$ recurrent neurons.

In this respect, also note that the output layer is applied to the final state computed by the recurrent component for each input sequence in all models. During the model selection, the nanowire-related hyparameters of the NW-ESN and NW-RNN models were set to the values reported in Table 2, which provided a set of physically meaningful values.

| Hyperparameter | Value |
|---|---|
| $K_{p0}$ | 0.0001 |
| $K_{d0}$ | 0.5 |
| $\eta_p$ | 10 |
| $\eta_d$ | 1 |

Table 2: Fundamental nanowire-related hyperparameters values used in the experiments.

| Hyperparameter | Values |
|---|---|
| input scaling $\omega$ | {1, 10} |
| bias scaling $\beta$ | {0, 0.001, 0.1, 1} |
| spectral radius $\rho$ | {0.8, 0.9, 0.95, 0.99} |
| $\gamma$ | {0.1, 0.5, 0.8, 0.95, 1} |
| sloe parameter $s$ | {1, 5} |
| discretization time $\Delta_t$ | {0.1, 0.01, 0.001} |
| N | {100} |

(a) NW-ESN.

| Hyperparameter | Values |
|---|---|
| epochs | {5000} |
| patience | {10} |
| learning rate | {0.001, 0.01} |
| batch size | {64, 128} |
| N | {100} |
| $\gamma$ | {0.5, 0.8, 0.95, 1} |
| slope parameter $s$ | {1, 5} |
| discretization time $\Delta_t$ | {0.01, 0.1} |

(b) NW-RNN.

| Hyperparameter | Values |
|---|---|
| input scaling $\omega$ | {1, 10} |
| bias scaling $\beta$ | {0.001, 0.1, 1} |
| spectral radius $\rho$ | {0.8, 0.9, 0.95, 0.99} |
| leaking-rate $\alpha$ | {0.1, 0.3, 0.5} |
| N | {100} |

(c) ESN.

| Hyperparameter | Values |
|---|---|
| epochs | {5000} |
| patience | {10} |
| learning rate | {0.001, 0.01} |
| batch size | {64, 128} |
| N | {100} |

(d) RNN.

Table 3: Values of the hyperparameters explored during model selection for the different models used in the experiments.

Tables 3c, 3a, 3d and 3b show the ranges of values explored for the remaining hyperparameters for NW-ESN, NW-RNN, ESN and RNN, respectively. After the model selection phase, the model evaluation of all models was based on the average performance over five trials.

## 4.1 Results

The results of our experiments, presented in Table 4, cover both classification and regression tasks. The scores for the classification tasks (FordA, ECG5000, SyntheticControl, Earthquakes) are expressed as accuracy ($\uparrow$), while the regression tasks (FloodModeling1 and FloodModeling2) are reported as root mean squared errors ($\downarrow$).

The results achieved demonstrate the competitive performance of nanowire-based models (NW-ESN and NW-RNN) in time-series tasks compared to standard methods such as ESN and RNN.

| Task | ESN | NW-ESN *(ours)* | RNN | NW-RNN *(ours)* |
|---|---|---|---|---|
| FordA($\uparrow$) | $0.544 \pm 0.017$ | $\mathbf{0.580 \pm 0.009}$ | $0.513 \pm 0.024$ | $0.515 \pm 0.000$ |
| ECG5000($\uparrow$) | $0.917 \pm 0.003$ | $0.921 \pm 0.002$ | $0.915 \pm 0.026$ | $\mathbf{0.931 \pm 0.003}$ |
| SyntheticControl($\uparrow$) | $0.875 \pm 0.009$ | $\mathbf{0.922 \pm 0.013}$ | $0.891 \pm 0.071$ | $0.891 \pm 0.016$ |
| Earthquakes($\uparrow$) | $\mathbf{0.763 \pm 0.008}$ | $0.755 \pm 0.006$ | $0.740 \pm 0.015$ | $0.748 \pm 0.000$ |
| FloodModeling1($\downarrow$) | $0.019 \pm 0.000$ | $\mathbf{0.008 \pm 0.000}$ | $0.023 \pm 0.002$ | $0.009 \pm 0.000$ |
| FloodModeling2($\downarrow$) | $0.018 \pm 0.000$ | $\mathbf{0.015 \pm 0.000}$ | $0.023 \pm 0.003$ | $0.017 \pm 0.000$ |

Table 4: Experimental results. Best results are highlighted in bold.

We can first observe that NW-RNN leads to a consistent improvement compared to the conventional RNN approach. This suggests that introducing nanowire-inspired dynamics into the fully trainable RNN model can yield benefits in complex time-series classification tasks, while still maintaining the flexibility of full backpropagation. However, what stands out most is the remarkable performance of the NW-ESN model. Notably, NW-ESN generally outperforms (with a single exception) the conventional ESN, sometimes by a significant margin. For instance, on the SyntheticControl dataset, NW-ESN shows a clear improvement in accuracy, highlighting the ability of the nanowire-based reservoir to suitably capture more complex temporal patterns. This improvement is especially noteworthy considering that NW-ESN retains the same computational complexity as ESN, as the reservoir dynamics are fixed and only the readout is trained.

Finally, observe that NW-ESN achieves the best overall performance on the majority of the tasks, despite having fewer trainable parameters compared to fully trainable models like NW-RNN. This efficiency, combined with its strong performance, emphasizes the potential of nanowire-based reservoirs as a powerful yet lightweight alternative for time-series processing, offering both scalability and accuracy without the added complexity of training recurrent dynamics.

## 5    Conclusions

In this paper, we introduced and explored the application of nanowire-based neural networks (NW-ESN and NW-RNN) for time-series processing, comparing their performance with traditional models such as ESNs and RNNs. Our experimental results demonstrate that nanowire-based models can offer competitive, and in many cases superior, performance.

NW-ESN, in particular, achieved the best performance in the majority of the tasks, highlighting the effectiveness of integrating the inherent nonlinear dynamics of nanowire networks into the Reservoir Computing paradigm. Despite having the same computational complexity as ESNs, NW-ESN exhibited significant improvements in performance across tasks, showcasing the potential of nanowire reservoirs to capture complex temporal patterns effectively. Moreover, NW-RNN demonstrated its capacity to enhance fully trainable models like RNNs by incorporating nanowire dynamics, leading to slight gains in classification and regression tasks. These results suggest that nanowire-based models offer a promising path toward the development of more efficient, adaptive, and scalable neural architectures for time-series analysis.

This work represents a step forward in the field of neuromorphic computing, underscoring the potential of nanowire-based architectures to bridge the gap between biological information processing and machine learning. Future research can build upon this foundation by further investigating the scalability of these models and extending their applications to more complex tasks and domains, ultimately advancing the capabilities of next-generation intelligent systems.

### Acknowledgments and Disclosure of Funding

This work has been supported by NEURONE, a project funded by the European Union - Next Generation EU, M4C1 CUP I53D23003600006, under program PRIN 2022 (prj. code 20229JRTZA, Italian Ministry of University and Research), and by and EU-EIC EMERGE (Grant No. 101070918).

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
