# OpenReview forum: "Nanowire Neural Networks for time-series processing"
_NeurIPS.cc/2024/Workshop/MLNCP — MLNCP Poster_

### Official Review · Reviewer_KMtw · 2024-09-27
**Relevant topic and method, but missing standards of evaluation set by the ML community**

**Rating:** 6
**Confidence:** 4

**Review:**

# Summary
The paper investigates a model of interactions of nanowires from a machine learning perspective. Recurrent neurons are modeled via a potentiation-depression rate balance equation that was published in 2020. A discrete time recurrent neural network formulation is obtained from the continuous neuron dynamics. Experiments are conducted on 4 classification and 2 regression datasets, all of which are small scale in terms of samples and sequence lengths. The authors present results for fully trained RNNs and reservoir approches where the recurrent dynamics are kept constant after random initialization, and only the linear read-out is trained.
# Discussion
The topic of the paper, namely using the physics of networks of nanowires for time-series modeling, aligns well with the Workshop.

Since the neuron model was already presented in Miranda et al. (2020), the contribution of this work is a study of the suitability of networks of these neurons for machine learning. Along these lines, the paper claims "competitive, and in many cases superior, performance". From a machine learning perspective, we would expect comparison and evaluation on standard benchmarks to back the claims made. Here, the reviewer sees major potential for improving the paper.
1. No comparison with related works: To convince the reader that the implementation is competitive, we need fair comparison with relevant published baseline models on relevant datasets. How does the presented model compare against previously published models?
2. Lack of standard benchmarks: The machine learning community has worked out many standardized benchmarks where dozens of baselines from other works exist to compare against. Some simple ones serve as sanity checks such as copy and memory tasks. Then there is a hierarchy of difficulty with increasing value for convincing the reader (non-exhaustive list: sequential MNIST, small-scale audio classification like Speech Commands, small-scale language modeling like Penn Treebank, of course larger scale audio/language tasks).
3. Hyperparameter search not extensive: RNNs can be quite sensitive to hyperparameters such as initialization and especially learning rates. The presented search does not convince the reviewer that one can conclude that the reservoir approach competes with fully trained RNNs (lines 133-137). E.g. all networks achieve basically random chance level (50%) on the FordA binary classification task, which might be evidence that the fully trained RNN was not able to learn properly.

Overall, the data presented is not sufficient to back the claims made. For example, we can be sure that fully trained networks are required to solve real problems like language modeling. To the best of my knowledge there is no competitive reservoir model that can compete with trained networks even on small scale language modeling (e.g. Penn Treebank), not even speaking of large-scale language modeling. The claim about "in many cases superior performance" (line 142) hence seems out of place.

An interesting observation of nanowire networks is that their state update is input dependent, a mechanism known as gating in the RNN community. Although the presented form is explicitly coded by the device physics instead of fully learned, the recent progress on gated RNNs such as Mamba and the likes, might serve as a motivation for further studying the presented model. For example, gated RNNs seem to admit stronger associative recall properties compared to linear-time invariant models.

The paper is technically correct. However, it requires more convincing evaluation on the path to a full paper. I recommend accepting it for this workshop if the claims made are weakened and set in context of the limited evaluation. Please also add a few lines to explain why the datasets were chosen, what we can learn from experiments on these datasets and why we can learn it from the data.
Further, I'd suggest to work on stronger evaluation for a full paper.
# Specific comments by line
- 29: Please clarify what is meant by "fault tolerance and adaptability" and provide evidence. e.g. spiking networks are often related with these attributes, but eventually are no more robust against e.g. adversarial examples than regular networks.
- 44: Figure 1 instead of 2?
- 63: Could you clarify why the state is 0<h<1 ? This is not immediately clear to me due to the exponential gating function, which might push the state to larger values?
- 79: Can the rescale operation be implemented in a physical nanowire network?
- 85-89: Can the authors discuss if these conditions can be met in manufacturing?
- 95-98: Datasets need a bit more description. It is neither clear nor convincing why these datasets are chosen and why well established datasets with established baseline models are ignored in this work.
- 133-137: As mentioned earlier, if an ESN performs as well as a trainable RNN there is either nothing to learn (i.e. the random features are fully descriptive of the task) or the RNN fails to learn. If there is nothing to learn, the task is too simple. I am absolutely sure that there is a scale at which random features will not suffice (and we probably don't have to move to GPT4 scale to see this). The presented evaluation methodology does not convince me that enough effort was spend to make the regular RNNs learn well enough to outperform the ESN. It rather seems that simple tasks and limited evaluation benefit the results.
- 142: Does the data back this claim?
- 143-145: does the data back this claim?
- 152: Overstating

# Limitations of the reviewer
The reviewer cannot evaluate the actually connections to manufacturing these systems, e.g.
- what is the precision of manufacturing?
- Is there drift in the parameters over longer times?
- Are the presented models robust against such physical processes?

---

### Official Review · Reviewer_aZaF · 2024-10-02
**Nanowire dynamics increase performance of time-series processing**

**Rating:** 8
**Confidence:** 4

**Review:**

Wow very interesting paper! The authors explore nanowire dynamics by creating the NW-ESN reservoir model and the NW-RNN fully trainable nanowire inspired RNN model. The motivation here is to see if these nanowires can be used for novel Neuromorphic architectures. The authors not only showed that they can match performance with their nanowire networks, but they actually showed it actually improved upon the conventional echo state network in reservoir computing and for the common RNN architectures. The paper is well written, thorough in the math behind the nanowire dynamics, and the authors conducted experiments on 6 different datasets in two tasks (classification and regression).


I just have one minor question:
- would you be able to expand on why the nanowire dynamics actually improved performance? Especially why does NW-ESN perform better than the RNN version when the RNN is fully learnable? Could the RNN case not have learned the NW-ESN case? The experiments are all there, but the paper would benefit from a little more insight for why the nanowire based models are so successful. Why are the able to 'capture more complex temporal patterns'?
- typos: L.44 -- fig. 1 instead of fig. 2 shows the photo.

Overall, very interesting paper

---

### Decision · Program_Chairs · 2024-10-10

Accept (Poster)